# Association of Tocilizumab and Invasive Aspergillosis in Critically Ill Patients with Severe COVID-19 Pneumonia and Acute Respiratory Distress Syndrome

**DOI:** 10.3390/jof8040339

**Published:** 2022-03-24

**Authors:** Kuo-Lun Wu, Chia-Yuan Chang, Heng-You Sung, Ting-Yu Hu, Li-Kuo Kuo

**Affiliations:** 1Division of Pulmonary and Critical Care Medicine, Mackay Memorial Hospital, Taipei City 10449, Taiwan; t123997889@gmail.com; 2Department of Internal Medicine, Mackay Memorial Hospital, Taipei City 10449, Taiwan; bluesky7952@gmail.com (C.-Y.C.); henrycgsh601@gmail.com (H.-Y.S.); 3Department of Critical Care Medicine, Mackay Memorial Hospital, Taipei City 10449, Taiwan; 4Department of Medicine, Mackay Medical College, New Taipei City 25245, Taiwan

**Keywords:** acute respiratory distress syndrome, COVID-19-associated pulmonary aspergillosis, tocilizumab

## Abstract

Coronavirus disease-2019 (COVID-19) causes severe pneumonia and acute respiratory distress syndrome. According to the current consensus, immunosuppressants, such as dexamethasone and anti-interleukin-6 receptor monoclonal antibodies, are therapeutic medications in the early stages of infection. However, in critically ill patients, viral, fungal, and bacterial coinfection results in higher mortality. We conducted a single-center, retrospective analysis of 29 mechanically ventilated patients with artificial airways. Patients were adults with confirmed COVID-19 infection and severe pneumonia. Acute respiratory distress syndrome was diagnosed according to the Kigali modification of the Berlin definition. Six patients had invasive pulmonary aspergillosis coinfection based on elevated serum galactomannan levels and/or bronchoalveolar lavage fluid. We present two cases with brief histories and available clinical data. We also conducted a literature review to determine whether immunosuppressants, such as tocilizumab, increase infection risk or invasive aspergillosis in patients with COVID-19. There is no conclusive evidence to suggest that tocilizumab increases coinfection risk. However, further studies are needed to determine the optimal dose, between-dose interval, and timing of tocilizumab administration in patients with COVID-19.

## 1. Introduction

Coronavirus disease-2019 (COVID-19), known as severe acute respiratory syndrome coronavirus 2 (SARS-CoV-2) infection, causes severe pneumonia and acute respiratory distress syndrome (ARDS) and is associated with a high mortality rate. As of 5 August 2021, there have been >5.4 million deaths and >290 million confirmed cases worldwide, according to the World Health Organization, with 850 deaths and >20,000 confirmed cases in Taiwan, according to the Central Epidemic Command Center of Taiwan. According to the current consensus, immunosuppressants (e.g., steroids [1], tocilizumab [2], and anti-interleukin-6 receptor monoclonal antibodies) play an important role in the early stages of infection. Among patients with severe pneumonia and ARDS, viral, fungal, and bacterial coinfection and secondary infection contribute to increased mortality [3]. Studies of *Aspergillus* coinfection among patients with COVID-19 are limited. Therefore, we conducted a single-center, retrospective analysis and literature review of cases of COVID-19 at Mackay Memorial Hospital in Taipei, Taiwan, between 19 May and 20 November 2021. Clinical, imaging, and laboratory data of 29 critically ill adult patients were collected and analyzed. Clinical course and outcomes were also evaluated.

## 2. Case Presentation

Since May 2021, more than 200 patients have been admitted to Mackay Memorial Hospital in Taipei, Taiwan. Standard treatment was performed according to consensus guidelines established by Taiwan Centers for Disease Control [4]. Thirty-two patients were admitted to the intensive care unit, and 29 of them had severe pneumonia requiring mechanical ventilation, coinciding with the definition of ARDS; one patient was admitted due to pregnancy requiring cesarean section, one patient was admitted due to massive vaginal bleeding with hemodynamic instability, and the last patient was admitted due to traumatic intracerebral hemorrhage with disturbance of consciousness. There was no construction or renovation during the pandemic in the care units. Blood samples were tested for galactomannan (*Aspergillus* antigen) twice weekly after patients were transferred to the intensive care unit (ICU) [5]. Due to the remarkable false-negative rate of serum galactomannan test, especially in cases of COVID-19-associated pulmonary aspergillosis (CAPA) [6], frequent blood tests were performed, and bronchoalveolar lavage galactomannan tests were performed for those with high clinical suspicion of invasive aspergillosis. Tocilizumab was administered to 26 out of the 29 patients. We reviewed the clinical courses of these patients. A brief history of two cases is presented. We also reviewed the literature on the association between tocilizumab and invasive aspergillosis. The study was conducted in accordance with the Declaration of Helsinki and approved by the Institutional Review Board of Mackay Memorial Hospital in Taipei, Taiwan (protocol code: 21MMHI407e; date of approval: 8 February 2022). A waiver for written informed consent was obtained.

Six patients who had received tocilizumab were diagnosed with invasive pulmonary aspergillosis during their ICU stay. The characteristics of these six patients are summarized in Table 1.

### 2.1. Representative Case 1 (Patient 3)

A 61-year-old man with a history of hypertension and type 2 diabetes mellitus presented to the emergency department (ED) with progressive dyspnea on exertion and hypoxia, having been under hotel quarantine. His oxygen saturation in room air was 85%. In the ED, bilateral opacities on chest radiography (Figure 1A) and symptom progression were observed. The test for COVID-19 was positive with an Xpert E gene cycle threshold value of 26.8. A non-rebreathing mask (15 L/min) was used to maintain oxygen saturation > 94%. Physical examination was significant for bilateral rales on auscultation of the lungs. On admission, intravenous ceftaroline and oral azithromycin were administered, along with remdesivir, dexamethasone, and tocilizumab (280 mg, 4 mg/kg (lower dose due to elevated liver enzymes)). The patient was transferred to the ICU with progressive hypoxemia (oxygen saturation 91% under a non-rebreathing mask (15 L/min)). He received high-flow nasal cannula (HFNC) and was switched to piperacillin–tazobactam and levofloxacin 3 days after admission to the ICU. The patient responded poorly to HFNC and was soon placed in the prone position. On Day 6, intubation with mechanical ventilation was performed for hypoxia, and the second dose of tocilizumab (4 mg/kg) was administered for suspected cytokine release syndrome (CRS) based on increased inflammatory markers [6] (lactate dehydrogenase (LDH): 418 IU/L; D-dimer: 809 ng/mL). Chest radiography revealed bilateral lung infiltration, indicating rapidly progressive diffuse pulmonary edema (Figure 1B).

On Day 7, positive end-expiratory pressure increased to 12 mmHg, with a fraction of inspired oxygen (FiO_2_) of >75% required to maintain oxygen saturation > 90%. The patient was placed in the prone position for 97 h. The blood *Aspergillus* antigen (galactomannan) titer on Day 15 was 0.73. The antifungal agent micafungin, used since Day 13, was switched to intravenous voriconazole [7]. Oxygenation and chest radiography improved, and the patient was extubated on Day 18. HFNC was well tolerated after extubation. The patient was transferred to the general ward for further treatment for ventilator-associated pneumonia. On Day 18, his sputum culture was positive for *Burkholderia cenocepacia*. Antibiotics and antifungal agents were discontinued 10 days later. The patient was discharged on Day 28.

### 2.2. Representative Case 2 (Patient 5)

A 61-year-old man with chronic obstructive pulmonary disease history was admitted to the ED with dyspnea on exertion. Bilateral opacities were observed on chest radiography (Figure 2A). The patient had remarkable hypoxemia with oxygen saturation of 82% in room air (96% under a simple mask (10 L/min)). Remdesivir, dexamethasone, and empiric antibiotics (intravenous ceftaroline and oral azithromycin) were administered on admission. On Day 3, tocilizumab (6.25 mg/kg) was administered under the suspicion of cytokine release syndrome (C-reactive protein: 13.5 mg/dL; normal range: 0–0.79; LDH: 413 IU/L). Progressive hypoxemia (oxygen saturation approximately 91% under a non-rebreathing mask (15 L/min)) was observed within a few hours. The patient was intubated and transferred to the ICU for further treatment. The patient was placed in the prone position for 23 h because of persistent hypoxemia and the need for extremely high FiO_2_ after intubation. On Day 6, chest radiography revealed patchy opacities in the right lower lung and bilateral subcutaneous emphysema (Figure 2B). A muscle relaxant (cisatracurium) was administered, and the ventilator settings were adjusted by reducing the positive end-expiratory pressure from 12 to 7 mmHg and the tidal volume from 400 to 350 mL. For newly developed patchy opacities, antibiotics were switched to cefoperazone-sulbactam and levofloxacin for better nosocomial infection control. On Day 7, due to increased oxygen demand (FiO_2_ titrated up from 50% to 100%), the patient was placed in the prone position for 46 h before being switched to the supine position on Day 8. In the “dry lung” strategy, furosemide (20 mg) was administered intravenously twice daily because of the increased body weight and stationary chest X-ray. On Day 10, an empiric antifungal agent (voriconazole) [7] was administered for persistent hypoxemia, with a PaO_2_/FiO_2_ ratio of approximately 120, despite the absence of fever episodes. Laboratory data showed leukocytosis (112,000/uL) with left shift (band: 3%; segment: 87%; lymphocytes: 4%), elevated LDH (291 IU/L), elevated D-dimer (654 ng/mL), and elevated ferritin (1040.90 ng/mL). Blood and sputum cultures were repeatedly negative. The patient responded poorly to chest X-rays and oxygenation. A second dose of tocilizumab (3.1 mg/kg) was administered, and the antibiotic regimen was switched to intravenous meropenem, linezolid, and colistin on Day 11. After three days of observation, hypoxemia progressed, and the patient was placed in the prone position for 52 h on Day 14. Umbilical cord-derived mesenchymal stem cells were administered on Days 14 and 17. Micafungin was administered as “add-on” therapy [7]. On Day 18, sputum cultures (collected on Day 15) were positive for *Ralstonia mannitolilytica*. The blood *Aspergillus* antigen titer on Day 14 was 0.76. The antifungal agent was switched to intravenous posaconazole due to blurred vision, one of the common adverse effects of voriconazole [7]. After seven days of treatment, oxygenation improved, and the patient was extubated on Day 26. HFNC was continued; however, severe orthodeoxia (oxygen desaturation 85% while in the sitting position) occurred. After a short period of rehabilitation, the patient was transferred to the general ward on Day 36. Improvement was observed on a chest X-ray (Figure 2C), and the patient was discharged on Day 61.

### 2.3. Patient Review

Of the 29 critically ill patients admitted to the ICU with severe pneumonia and ARDS who tested positive more than twice for SARS-CoV-2 by real-time reverse transcriptase-polymerase chain reaction, six had invasive pulmonary aspergillosis coinfection based on elevated serum galactomannan levels and/or bronchoalveolar lavage fluid. Among them, two patients died and four patients were discharged. Five patients presented with comorbid type 2 diabetes mellitus, and one patient had a history of chronic obstructive pulmonary disease. All six patients received a full 10-day course of anti-inflammatory treatment with dexamethasone and at least one dose of tocilizumab (8–15 mg/kg). Adequate antifungal treatment was prescribed for all patients with probable invasive aspergillosis [7]. Additional clinical characteristics, including symptoms and initial performance at presentation, are provided in Table 1. Tocilizumab administration, diagnostic time, treatment of invasive aspergillosis, and outcomes are shown in Table 2.

## 3. Discussion

COVID-19 pathophysiology suggests that the middle stage of infection is associated with the strongest inflammatory response, which may lead to CRS and ARDS, while the later stages are associated with an increased risk of secondary infection, including viral, bacterial, and fungal infection, and death [8]. The ideal time frame for immunosuppressants, such as corticosteroids and interleukin-6 antagonists, may be in the early-to-mid stage, after viral replication and before the inflammatory response. If immunosuppressants are administered in the later stages of infection, they may lead to more severe secondary infections and potentially harm the patient.

Previous studies reported that the incidence of CAPA was 7% among critical patients in Austria [9], 7.6% among critical patients in Wales, UK [10], and 2.2% among mortality cases in Taipei, Taiwan [11]; a lack of close, routine monitoring of galactomannan levels was noticed, which might lead to underestimating the incidence of CAPA. Nevertheless, due to the relatively high incidence of CAPA (6/29) compared with these studies, all 29 patients were reviewed with no specific environmental exposures or contributory occupational history. Environmental monitoring of the ICU was also performed twice monthly to check for any contamination, and the presence of SARS-CoV-2, multiple-drug resistant organisms, or fungi was not detected.

A recent study [12] showed that COVID-19 is associated with a higher risk of CAPA. The mortality rate of COVID-19 is estimated to be 1–6% in the general population. In our observational review, the mortality rate of invasive aspergillosis (33%, two out of six) was similar to that of patients with invasive pulmonary aspergillosis in the general population (30.22%) according to prior studies in Taiwan [13] and also similar to that of patients with CAPA (31.4%) in Spain [14].

It is reported that prior administration of certain antibiotics, especially penicillin, might raise the false-positive rate of the serum galactomannan test [15]. In this study, two of our reported six patients received piperacillin–tazobactam for a few days (8 days, 3 days) before being diagnosed with invasive aspergillosis. After discontinuing piperacillin/tazobactam, serum *Aspergillus* Ag titer remained high for 3 weeks in the two cases. Thus, the false-positive rate of the serum galactomannan test was low, and both cases survived the infection.

Studies [16,17,18,19] have shown that tocilizumab, which is used to treat severe or critically ill patients with COVID-19, is associated with a higher risk of secondary infections, especially bacterial infections. Moreover, it is also reported that the prescription of tocilizumab was one of the independent risk factors associated with CAPA in Spain [14]. However, another study [8] reported no significant differences in infection rates between the tocilizumab and non-tocilizumab groups.

In a meta-analysis [20] of the relationship between the administration of interleukin-6 antagonists and mortality among hospitalized patients with COVID-19, a combination of interleukin-6 antagonists and corticosteroids was associated with lower 28-day mortality than corticosteroids or interleukin-6 antagonists alone. No significant differences in 28-day secondary infection rates were observed among those treated with interleukin-6 antagonists, corticosteroids, or a combination of the two. However, a case-based review by Cai et al. [21] mentioned that prescribed glucocorticoids might delay viral clearance, raising a concern of COVID-19 recurrence.

Another study [22] reported a statistically significant increase in the length of ICU stay (3.73 vs. 1.91; *p* < 0.01), duration of mechanical ventilation (3.58% vs. 1.95%; *p* < 0.01), and duration of hospitalization (3.83% vs. 2.58%; *p* < 0.01) in patients who received tocilizumab and developed a fungal infection compared with those who were treated with standard COVID-19 therapies.

In a review article [23], CAPA and colonization with *Aspergillus* spp. were analyzed separately. Patients with CAPA had a higher rate of tocilizumab administration (100% vs. 40%; *p* = 0.04) and a longer period from COVID-19 detection to confirm the diagnosis of aspergillosis (median days, 15 vs. 3; *p* = 0.008).

Regarding whether the accumulative dose or multiple doses of tocilizumab confers any additional benefit in patients with COVID-19, a strong dose-response relationship was observed. In a retrospective observational study [22], the fungal infection rate was 2.5% in patients who did not receive tocilizumab, 20% in those who received a single dose (RR, 17.5%; *p* < 0.01), and 50% in those who received two doses (RR, 47.5%; *p* < 0.01). Another retrospective study [24] evaluated all-cause mortality, the rate of ICU admission, the rate of intubation, the incidence of septic shock, and the duration of hospital stay in patients who received an additional dose of tocilizumab. The authors concluded that there was no additional benefit of multiple doses of tocilizumab in preventing all-cause mortality, septic shock, and secondary infections.

## 4. Conclusions

In patients who receive immunosuppressants such as interleukin-6 antagonists or glucocorticoids, the possibility of co-infection with bacteria, fungus, or virus is likely. As mentioned above, close monitoring of serum galactomannan levels and bronchoalveolar lavage is recommended for early detection and initiation treatment of occult fungal infection. There is no consensus on the association between tocilizumab and the secondary fungal infection rate, including the invasive aspergillosis infection rate. Further studies are needed to determine the optimal dose, between-dose interval, and timing of tocilizumab administration in patients with COVID-19.

## Figures and Tables

**Figure 1 jof-08-00339-f001:**
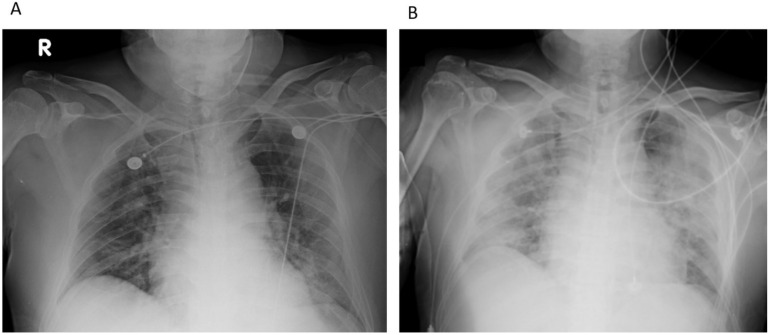
Representative case 1 (Patient 3): (**A**) Bilateral opacities on chest radiography taken in the emergency department. (**B**) Chest radiography revealed bilateral lung infiltration, indicating rapidly progressive diffuse pulmonary edema.

**Figure 2 jof-08-00339-f002:**
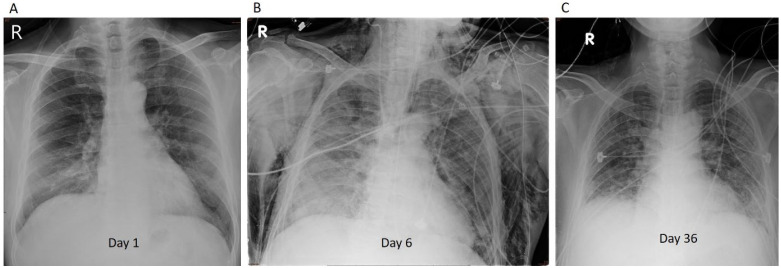
Representative case 2 (Patient 5): (**A**) Bilateral opacities were observed on chest radiography taken in emergency department. (**B**) On Day 6, chest radiography revealed patchy opacities in the right lower lung and bilateral subcutaneous emphysema. (**C**) On Day 36, improvement was observed on chest X-ray.

**Table 1 jof-08-00339-t001:** Clinical characteristics and initial presentations of six patients with COVID-19 complicated by invasive aspergillosis who received tocilizumab.

	Patient 1	Patient 2	Patient 3	Patient 4	Patient 5	Patient 6
Age	72	53	61	63	61	77
Sex	Male	Male	Male	Male	Male	Female
Weight (kg)	77	73	70	53	64	66
Tobacco exposure	Current smoker	Never smoked	Current smoker	Never smoked	Former smoker	Never smoked
Comorbidities	DM, HbA1c: 6.6% Wegner granulomatosis	DM, HbA1c: 9.2%	DM, HbA1c: 11% HTN	DM, HbA1c: 9.4% HTN	COPD	DM, HbA1c: 6.4%
COVID-19 symptoms	Dyspnea, cough	Cough, myalgia	Cough, fever, dyspnea	Cough, fever, diarrhea	Fever, dyspnea, cough	Cough, dyspnea
Time from symptom onset to diagnosis (day(s))	3	2	4	1	1	1
APACHE II scores	23	11	13	26	18	31

Note: DM: diabetes mellitus; HbA1c: Hemoglobin A1c; COPD, chronic obstructive pulmonary disease; HTN: hypertension; APACHE II, Acute Physiology and Chronic Health Evaluation II.

**Table 2 jof-08-00339-t002:** Tocilizumab administration, diagnostic time, treatment of invasive aspergillosis, and outcomes.

	Patient 1	Patient 2	Patient 3	Patient 4	Patient 5	Patient 6
First tocilizumab dose	5.2 mg/kg	11 mg/kg	4 mg/kg	7.5 mg/kg	6.25 mg/kg	6 mg/kg
Second tocilizumab dose	5.2 mg/kg	n/a	4 mg/kg	7.5 mg/kg	3.1 mg/kg	6 mg/kg
Days between 1ST and 2nd tocilizumab dose	3	n/a	4	6	8	11
Total cumulative tocilizumab dose	10.4 mg/kg	11 mg/kg	8 mg/kg	15 mg/kg	9.38 mg/kg	12.12 mg/kg
Status of capa ^$^	Probable	Probable	Probable	Probable	Probable	Probable
Usage of penicillin before diagnosis	No	No	Yes, piperacillin/tazobactam	Yes, piperacillin/tazobactam	No	No
Blood *Aspergillus* AG titer while IA diagnosed	0.69	0.61	0.73	1.82	0.76	1.09
The highest blood *ASPERGILLUS* AG titer	1.60	6.62	1.14	3.17	2.26	4.71
The highest BAL *ASPERGILLUS* AG titer	7.09	0.89	0.36	0.76	0.26	0.41
Imagine evidence of aspergillosis in chest CT	N/A	Nonspecific findings	N/A	Clusters of fluffy nodules	Diffuse ground glass opacity with reticulations	Diffuse ground glass opacity with consolidations
Days after COVID-19 symptom onset to IA diagnosed	32	32	15	21	20	27
Days after ICU admission to IA diagnosed	28	19	6	18	11	10
Days after 1ST tocilizumab dose to IA diagnosed	28	26	9	13	11	21
Days after 2nd tocilizumab dose to IA diagnosed	25	n/a	5	7	3	10
Antifungal agent	Voriconazole	Voriconazole + caspofungin, switched to posaconazole	Voriconazole	Voriconazole, add-on caspofungin	Voriconazole + micafungin, switched to posaconazole	Voriconazole, add-on caspofungin
Outcomes	Death on Day 51	Death on Day 168	Discharged on Day 28	Discharged on Day 76	Discharged on Day 61	Discharged on Day 151 (respiratory care ward)

Note: n/a: not applicable; IA: invasive aspergillosis; Ag: antigen; ICU: Intensive Care Unit; CAPA: COVID-19-associated pulmonary aspergillosis; BAL: bronchoalveolar lavage; CT: computed tomography. ^$^: according to Koehler 2020, Lancet Infect Dis [7].

## Data Availability

Not applicable.

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
