# Peer review of "Association of Tocilizumab and Invasive Aspergillosis in Critically Ill Patients with Severe COVID-19 Pneumonia and Acute Respiratory Distress Syndrome"

_jof, 2022, doi:10.3390/jof8040339_

Round 1

Reviewer 1 Report

The manuscript entitled “The Association of Tocilizumab and Invasive Aspergillosis in Critically Ill Patients with Severe COVID-19 Pneumonia and Acute Respiratory Distress Syndrome” describes very interesting scientific issue. The results provide an interesting added value in our comprehension whether immunosuppressants, such as tocilizumab, may increase the risk of infection or invasive aspergillosis in patients with COVID-19, and justify of the publication of this manuscript in the Journal of Fungi.

Some minor comments:

  1. Line 163-164 “The mortality rate of COVID-19 is estimated to be 1‒6% in the general population. In this observational review, the mortality rate of invasive aspergillosis was higher than that in the general population (17%).” Could you please compare those data with the mortality rate of invasive aspergillosis in the general population. What is the level of difference?
  2. The “Discussion” section should be also supplemented by the results presented below:

Cai S, Sun W, Li M, Dong L. A complex COVID-19 case with rheumatoid arthritis treated with tocilizumab. Clin Rheumatol. 2020 Sep;39(9):2797-2802. doi: 10.1007/s10067-020-05234-w. Epub 2020 Jun 19. PMID: 32562070; PMCID: PMC7303585.

Machado M, Valerio M, Álvarez-Uría A, Olmedo M, Veintimilla C, Padilla B, De la Villa S, Guinea J, Escribano P, Ruiz-Serrano MJ, Reigadas E, Alonso R, Guerrero JE, Hortal J, Bouza E, Muñoz P; COVID-19 Study Group. Invasive pulmonary aspergillosis in the COVID-19 era: An expected new entity. Mycoses. 2021 Feb;64(2):132-143. doi: 10.1111/myc.13213. Epub 2020 Nov 29. PMID: 33210776; PMCID: PMC7753705.

3. The conclusions section is very general, I am asking for more in-depth conclusions.

Author Response

# Comment 1.: Line 163-164 “The mortality rate of COVID-19 is estimated to be 1‒6% in the general population. In this observational review, the mortality rate of invasive aspergillosis was higher than that in the general population (17%).” Could you please compare those data with the mortality rate of invasive aspergillosis in the general population? What is the level of difference?

Response for comment 1.:

    Thank you for your suggestion.

    We have supplemented our discussion in the first and second paragraphs of the “Discussion” section.

In our observational review, the mortality rate of invasive aspergillosis (33%, 2 out of 6) was similar to that of patients with invasive pulmonary aspergillosis in the general population (30.22%) according to prior studies [9]

 We have relied on the study by Sun et al. (reference 9: Sun, K.S.; Tsai, C.F.; Chen, S.C.C.; Huang, W.C. Clinical outcome and prognostic factors associated with invasive pulmonary aspergillosis: an 11-year follow-up report from Taiwan. PLoS One 2017, 12, e0186422).

# Comment 2.:  The “Discussion” section should be also supplemented by the results presented below:

Cai S, Sun W, Li M, Dong L. A complex COVID-19 case with rheumatoid arthritis treated with tocilizumab. Clin Rheumatol. 2020 Sep;39(9):2797-2802. doi: 10.1007/s10067-020-05234-w. Epub 2020 Jun 19. PMID: 32562070; PMCID: PMC7303585.

Machado M, Valerio M, Álvarez-Uría A, Olmedo M, Veintimilla C, Padilla B, De la Villa S, Guinea J, Escribano P, Ruiz-Serrano MJ, Reigadas E, Alonso R, Guerrero JE, Hortal J, Bouza E, Muñoz P; COVID-19 Study Group. Invasive pulmonary aspergillosis in the COVID-19 era: An expected new entity. Mycoses. 2021 Feb;64(2):132-143. doi: 10.1111/myc.13213. Epub 2020 Nov 29. PMID: 33210776; PMCID: PMC7753705.

Response for comment 2.:

   Thank you for your suggestion.

   We have supplemented our discussion in the fourth and fifth paragraphs of the “Discussion” section.

# Comment 3:  . The conclusions section is very general, I am asking for more in-depth conclusions.

Response for comment 3.:

   Thank you for your suggestion.

   We have revised the “Conclusion” section to make it more meaningful.

Reviewer 2 Report

Line 16/17 “.., play an important role in the early stages of infection..” suggest “.., are therapeutic interventions  in the early stages of infection..”

Line 49/50. How many of these >200 patients were admitted to ICU?

Were either patient smokers?

Diagnosis appears to be based solely on the blood Aspergillus antigen titer. This is a limitation, surely. Did any have CT scan imaging of the chest?

Were there any environmental exposures? Construction, renovations?

Author Response

# Comment 1.: Line 16/17 “.., play an important role in the early stages of infection..” suggest “.., are therapeutic interventions  in the early stages of infection..”

Response for comment 1. :

    Thank you for your suggestion.

    We have made the minor correction in lines 19–20.

# Comment 2.: Line 49/50. How many of these >200 patients were admitted to ICU?

Response for comment 2. :

Thank you for your comment.

Thirty-two patients were admitted to ICU with confirmed COVID-19. However, 3 of them were excluded in this study due to:

1.    The first patient was admitted due to pregnancy requiring cesarean section. She was transferred to the general isolation ward 2 days after the operation.

2.    The second patient was admitted due to massive vaginal bleeding. She was transferred to the general ward on the second day due to cessation of active bleeding and stable hemodynamic status.

3.    The third patient was admitted due to traumatic intracerebral hemorrhage. Endotracheal intubation with mechanical ventilation was performed due to disturbance of consciousness but not due to respiratory failure.

    Since May 2021, >200 patients have been admitted to Mackay Memorial Hospital in Taipei, Taiwan, including ward and ICU. Overall, 29 patients were admitted to the ICU due to respiratory failure or severe pneumonia.

# Comment 3.:  Were either patient smokers?

Response for comment 3. :

Thank you for your suggestion.

We have added the data to table 1. There were two current smokers, one former smoker, and three non-smokers.

# Comment 4.:. Diagnosis appears to be based solely on the blood Aspergillus antigen titer. This is a limitation, surely. Did any have CT scan imaging of the chest?

Response for comment 4. :

Thank you for your suggestion.

    Due to the remarkable false negative rate of serum galactomannan test, especially in cases of COVID-19-associated pulmonary aspergillosis (CAPA) [6], frequent blood tests were performed, and bronchoalveolar lavage galactomannan tests were performed for those with high clinical suspicion of invasive aspergillosis.

    The results of BAL highest Aspergillus Ag titer, status of CAPA, and chest CT image evidence of aspergillosis have been added to Table 2.

# Comment 5.:  Were there any environmental exposures? Construction, renovations?

Response for comment 5. :

    Thank you for your suggestion.

    No specific environmental exposures or contributory occupational history was recorded.

Reviewer 3 Report

Introduction

Line 33: The disease is not also called SARS-CoV-2, the infecting virus is called SARS-CoV-2.

Case presentation

Line 50: Which consensus guidelines? Define or cite them.

Line 53: Serum galactomannan is a sensitive biomarker in hematological patients. For CAPA they are often negative even in proven cases. This should be mentioned. Please see Koehler 2020, Lancet Infect Dis.

Patient review

Line 148: I would propose to add for the positive BAL galactomannan levels in Table 2.

Please add the status possible, probable or proven diagnosis of CAPA to the patients you present according to Koehler 2020, Lancet Infect Dis.

Discussion

Please discuss that penicillin use can lead to false positive galactomannan results. Please discuss how many of the 6 patients with potential CAPA received penicillin before diagnosis.

Author Response

# Comment 1.: Line 33: The disease is not also called SARS-CoV-2, the infecting virus is called SARS-CoV-2.

Response for comment 1. :

    Thank you for your suggestion.  

We have corrected the terminology in line 43.

# Comment 2.: line 50: Which consensus guidelines? Define or cite them.

Response for comment 2. :

   Thank you for your suggestion.  

   We are referring to the guideline released by Taiwan’s CDC government: “新型冠狀病毒(SARS-CoV-2)感染臨床處置暫行指引, established by Taiwan CDC.”

# Comment 3.:   Line 53: Serum galactomannan is a sensitive biomarker in hematological patients. For CAPA they are often negative even in proven cases. This should be mentioned. Please see Koehler 2020, Lancet Infect Dis.

Response for comment 3. :

    Thank you for your suggestion.

    Due to the remarkable false negative rate of serum galactomannan test, especially in cases of COVID-19-associated pulmonary aspergillosis (CAPA) [6], frequent blood tests were performed, and bronchoalveolar lavage galactomannan tests were performed for those with high clinical suspicion of invasive aspergillosis. We relied on the findings by Koeler et al. published in 2020 Lancet Infect Dis (reference 6).

    The results of BAL highest Aspergillus Ag titer, status of CAPA, and chest CT image evidence of aspergillosis have been added to Table 2.

# Comment 4.:   Line 148: I would propose to add for the positive BAL galactomannan levels in Table 2.

Response for comment 4. :

    As you have suggested, we have added the results to Table 2.

# Comment 5.:   Please add the status possible, probable or proven diagnosis of CAPA to the patients you present according to Koehler 2020, Lancet Infect Dis.

Response for comment 5. :

    We have added the results to Table 2.

# Comment 6.:  Please discuss that penicillin use can lead to false positive galactomannan results. Please discuss how many of the 6 patients with potential CAPA received penicillin before diagnosis.

Response for comment 6. :

    Thank you for your suggestion.

    We have supplemented our discussion in the second paragraph of the “Discussion” section, mentioned in line 239-249.

    Data on the “usage of penicillin before diagnosis” were added to Table 2.
